# Prevalence and comorbidity of anxiety disorder in school-attending children and adolescents aged 6–16 years in China

Fang Wang,[1] Hanxue Yang,[2] Fenghua Li,[3] Yi Zheng,[4] Hui Xu,[5] Rui Wang,[6] Ying Li,[6] Yonghua Cui  [1]

YL and YC contributed equally.

YL and YC are joint senior authors.

For numbered affiliations see end of article.

**Correspondence to**
Professor Yonghua Cui; cuiyonghua@bch.com.cn

Dr Ying Li; yyjnly@126.com

## ABSTRACT

**Background** The reported prevalence rate of anxiety disorder in the paediatric population varies widely between different counties. Currently, there is no national epidemiological survey *of childhood* anxiety disorder in China. This study aims to investigate the national prevalence of anxiety disorder, the distribution profiles of different subtypes and its comorbidity rates among school students.

**Methods** A nationwide epidemiological survey of mental disorders in school students aged 6–16 years was conducted. Participants were randomly recruited from five provinces in China. The Child Behavior Checklist was used to screen students at high risk for mental disorders. The final diagnosis was made based on the Diagnostic and Statistical Manual-IV. The point prevalence and comorbidity rate of anxiety disorder were calculated, and the difference between age, sex and socioeconomic status groups was also compared.

**Results** Generalised anxiety disorder (GAD) was the most common anxiety disorder in school-attending children and adolescents, with a prevalence rate of 1.3% (95% CI: 1.2 to 1.3). Separation anxiety and specific phobia were more common in children than in adolescents. Girls had a higher prevalence of panic disorder (0.3% vs 0.2%, $\chi^2$=14.6, p<0.001) and agoraphobia (0.9% vs 0.8%, $\chi^2$=4.3, p=0.03) than that of boys. We found no significant difference between developed and less developed areas. Girls were more likely to have panic disorder and GAD than boys, with ratios of 2.13:1 and 1.01:1, respectively. The co-occurrence of anxiety disorder and attention-deficit and disruptive behaviour disorder was very common, ranging from 40% to 85%.

**Conclusions** Anxiety disorder was prevalent among school students in China, and comorbidity with attention-deficit and disruptive disorder was very common. The data imply that screening for anxiety disorder is needed in school settings. Policies should be adapted to provide psychological services to children and adolescents. A comprehensive assessment is recommended in clinical practice.

## WHAT IS ALREADY KNOWN ON THIS TOPIC

⇒ Anxiety disorder is common in the paediatric population. It will cause great distress to both children and their families. However, as an internal symptom, anxiety may be neglected by caregivers or teachers. This would prevent children and adolescents with anxiety disorders from accessing medical services. Currently, there are no nationwide epidemiological data on anxiety disorders in the paediatric population in China.

## WHAT THIS STUDY ADDS

⇒ This prevalence rate of anxiety disorder among school students in China was 4.7%. GAD was the most common subtype; the prevalence rate was 1.3%. Adolescents had higher rates of panic disorder, GAD, agoraphobia and social phobia than that of children. Girls had slightly higher rates of panic disorder, GAD and agoraphobia than that of boys. The comorbidity of attention-deficit and disruptive behaviour disorder and anxiety disorder is very common. The prevalence of anxiety disorder was not influenced by socioeconomic conditions.

## HOW THIS STUDY MIGHT AFFECT RESEARCH, PRACTICE OR POLICY

⇒ The data indicate screening for anxiety disorder may be needed in school settings, and policies should be adapted to provide psychological services to children and adolescents.

## INTRODUCTION

Anxiety disorders are characterised by excessive anxiety, worry and related behaviours. Anxiety disorders, including panic disorder, generalised anxiety disorder (GAD), specific phobia, separation anxiety disorder, specific disorder and agoraphobia, and social phobia, are common in the paediatric population.[1 2] Comorbidity with other mental disorders, such as major depressive disorder and substance abuse, is very common in anxiety disorder.[3] Anxiety disorder and its comorbidities not only cause great distress to children, adolescents and their families but also have a

negative impact on their educational achievement and social function in adulthood.[3 4] Furthermore, anxiety disorders increase the risk of other mental disorders.[5] Children with anxiety disorder are more likely to have depression disorder, suicidal behaviour and other psychiatric disorders in adulthood.[6] As anxiety is an internal symptom, caregivers and teachers may not recognise the onset of anxiety in children and adolescents, which would prevent children and adolescents with anxiety disorders from accessing medical services.[7] It would therefore be of great importance to know the prevalence and characteristics of anxiety disorder in school settings in order to adapt our policies and provide psychological services to children and adolescents.

There have been many surveys exploring the prevalence of anxiety disorder in the paediatric population. However, the results vary widely between countries, ranging from 2% to 31%.[3 8–11] The prevalence of anxiety disorder was 6.3% in children aged 7–10 years and 4% in adolescents aged 11–17 years in a German national survey including 17 641 participants.[8] A national survey in the USA including 10 123 adolescents reported that the lifetime prevalence was as high as 31.9%.[11] Another recent study reported a lifetime prevalence of 15.6% among Austrian adolescents.[10] Comorbidity with other mental disorders was also very common in anxiety disorder, and the results were also inconsistent. The coexistence of different subtypes of anxiety disorder was 17%–60%.[9 12] The comorbid rate of anxiety disorder and depression disorder was reported to be 10%–40%.[13] The co-occurrence rates of anxiety disorder and attention-deficit/hyperactivity disorder (ADHD) were similar in different studies. The comorbid rate was about 17%–25%.[9 14] The prevalence of conduct disorder (CD) was also high in patients with an anxiety disorder (18%–38%).[9 15]

At present, the national prevalence rate of anxiety disorders in the paediatric population in China is still not clear as there is no nationwide epidemiological survey. There are only a few studies conducted locally. A survey conducted in Sichuan province, including 20 752 6–16-year-old students, reported that the 6-month prevalence rate of anxiety disorder was 3.94%.[16] Chen *et al* investigated 10 118 children aged 7–12 years from primary and middle schools in Taiwan and reported that the lifetime prevalence of anxiety disorder was 15.2%.[17] The prevalence of any anxiety disorder among adolescents (grade 7–9 high school students, the mean age was 13.8±1.2 years) in Hong Kong was estimated to be 30.2%.[18] As you can see, the results vary a lot and cannot represent the overall situation in China. It is of great importance to estimate the national prevalence rate of paediatric anxiety disorder as it will offer data support for government officials and mental health service providers. In this study, we further analysed the data collected in the first national prevalence study of mental disorders in school-aged children and adolescents in China.[19] The aim was to investigate the prevalence of anxiety disorder subtypes, their comorbid conditions and the distribution profile between different ages, genders and locations.

## METHOD
### Included participants
This is a two-stage survey carried out in primary and high schools in Beijing, Liaoning, Jiangsu, Sichuan and Hunan provinces. The age range of school students is from 6 years to 17 years. Due to the consideration of the College Entrance Examination, we decided not to include students in the final year of secondary school. So, the age range of students included in this survey was 6–16 years. The five provincial administrative regions were chosen by taking the geographical partition, economic development and rural/urban factors into consideration. In the official document (National Bureau of Statistics of China, 2011), China has divided the country into northeastern, eastern, middle and western regions geographically. Liaoning is chosen as the representative of northeast area, Jiangsu as the representative of eastern area, Sichuan as the representative of western area and Hunan as the representative of southern area. Beijing was picked as being the representative of a Developed Urban Area, which had a gross domestic product per capita of US$12 736 and a population larger than 10 million. Then, we randomly selected 2–4 prefecture divisions in these five areas. 15 prefecture divisions in total were selected finally. Next, in each prefectural division, simple sampling without replacement was used to select schools one by one (with a predetermined ratio of primary schools to middle schools as 1:1). Then, we randomly selected 2–5 classes in each grade of every school. The sample size was calculated using the formula $n = \frac{z^2_{1-\frac{\alpha}{2}}(1-p)}{p\varepsilon^2}$.[19 20] We chose the prevalence of Tourette disorder (0.30%) to estimate the sample size as it was reported to be the lowest among children and adolescents aged 6–17 years[21]; with a confidence coefficient of 95% (Z) relative error (ε) of 15%, 56 742 participants would reach a statistic power of 1. We assumed 20% would be lost during the follow-up and decided to recruit 73 992 participants. Based on this, we randomly chose 1764 classes from 169 schools (81 primary schools and 88 high schools). A more detailed description of the study design was described in our previous study.[19]

### CBCL screening
In the first stage, the Child Behavior Checklist (CBCL) was used for screening. The Chinese version of the CBCL has been shown to have good reliability and validity in clinical use.[22] And the internalising syndrome subscale of CBCL had been proved to have good discriminant validity for anxiety disorder.[23] For all the enrolled students, their parents or other caregivers were asked to complete the CBCL, and the total problem score was calculated. According to the previous study of Liu *et al*, we chose a cut-off score of 35, which was the 90th percentile of the

CBCL Total Problems score in their study.[24] Students with CBCL Total Problems scores higher than 35 (including 35) were identified as high risk and underwent the second-stage interview.

## MINI-KID and DSM-IV interview

This stage was performed approximately 2 weeks after the completion of the first stage. Considering the false-negative rate of CBCL screening, we also randomly selected 5% of the participants who showed CBCL-negative results and included them in the stage 2 assessment. They and the high-risk individuals underwent detailed interviews using the Mini-International Neuropsychiatric Interview for Children and Adolescents (MINI-KID). The MINI-KID is a widely used structured interview tool for the diagnosis of psychiatric disorders in children and adolescents.[25] In our study, the student and his or her parent or caregiver were interviewed at the same time. The MINI-KID assessment was carried out by psychiatrists with an attending physician title or higher, who were trained in the basic

skill of MINI-KID. For estimation of the intraclass correlation coefficient (ICC) value, interview video clips of 10 children were watched and the children were rated. The ICC value was calculated based on the rating results of all the psychiatrists. The ICC value in our study was over 0.85. They also underwent the Diagnostic and Statistical Manual-IV (DSM-IV) interview. Psychiatrists who performed the DSM-IV interview referred to the results of the MINI-KID rating, but the final diagnosis was made based on the DSM-IV criteria. The study procedure is illustrated in figure 1.

## Statistical analysis

R software 3.4.3 was used for statistical analysis. The point prevalence of different subtypes of anxiety disorder and their comorbidity rates were calculated. The prevalence was adjusted based on the weights of their provincial region, prefectural division, county/district, school and class. Taylor series linearisation was used to calculate the 95% CIs of the prevalence rate. The locations included in

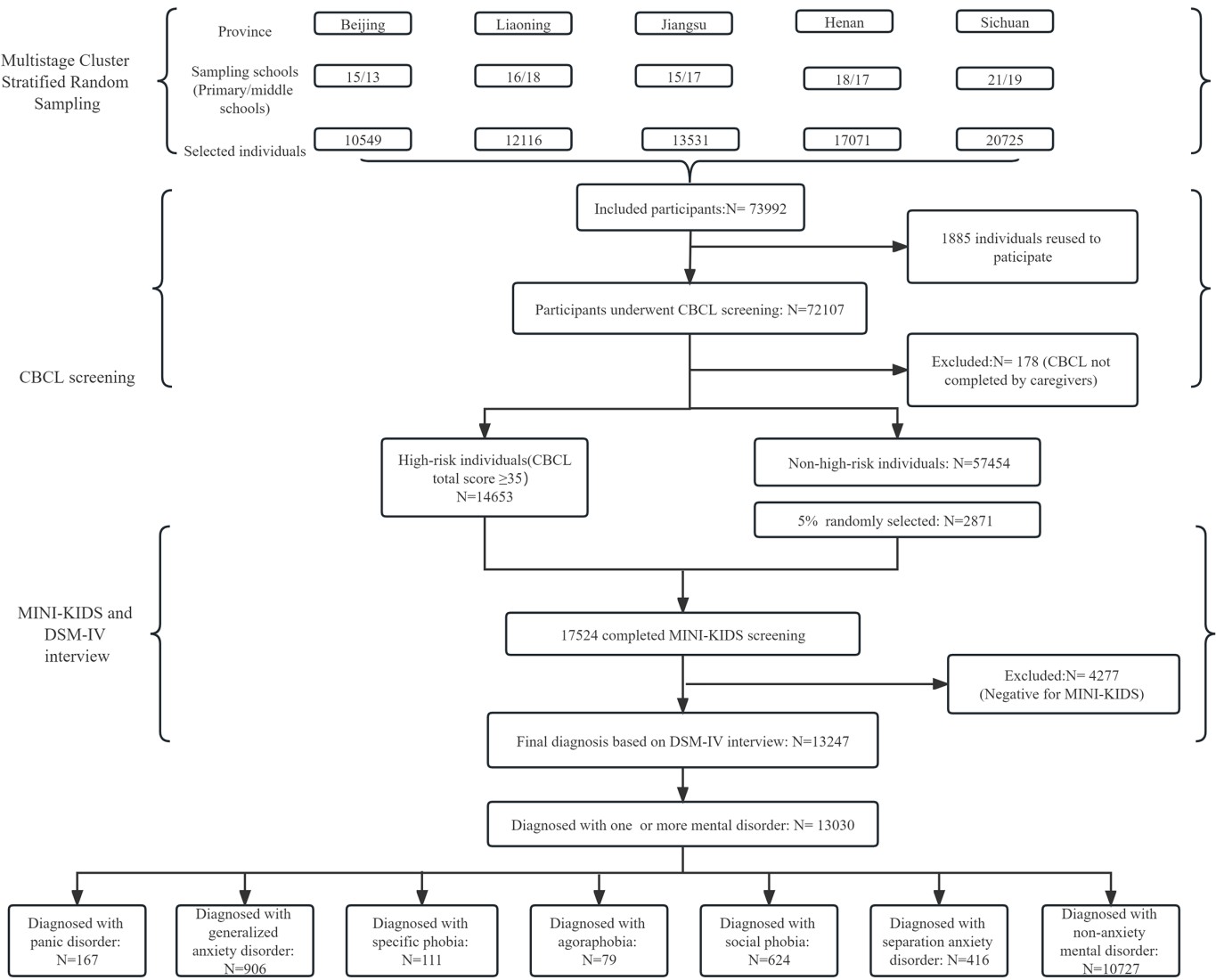

**Figure 1** The flowchart of the study. CBCL, Child Behavior Checklist; MINI-KID, Mini-International Neuropsychiatric Interview for Children and Adolescents; DSM-IV, Diagnostic and Statistical Manual-IV.

**Table 1** Point prevalence of anxiety disorder in Chinese school children and adolescents

| | Identified individuals | Unweighted point prevalence (%) | Unweighted 95% CI (%) | Adjusted point prevalence (%) | Adjusted 95% CI (%) |
|---|---|---|---|---|---|
| Generalised anxiety disorder | 906 | 1.3 | 1.2 to 1.3 | 1.3 | 1.2 to 1.3 |
| Social phobia | 624 | 0.9 | 0.8 to 0.9 | 0.8 | 0.8 to 1.0 |
| Separation anxiety disorder | 416 | 0.6 | 0.5 to 0.6 | 0.6 | 0.5 to 1.0 |
| Specific phobia | 111 | 0.2 | 0.1 to 0.2 | 0.2 | 0.1 to 0.2 |
| Panic disorder | 167 | 0.2 | 0.2 to 0.3 | 0.2 | 0.2 to 0.2 |
| Agoraphobia without panic | 79 | 0.1 | 0.1 to 0.1 | 0.1 | 0.1 to 0.1 |

this study were classified into developed areas (Beijing) and developing areas (Liaoning, Jiangsu, Sichuan and Hunan provinces) based on economic development conditions. The prevalence of anxiety disorder was also compared between age, gender and socioeconomic condition groups using Rao-Scott adjusted $\chi^2$ tests.[26] All statistical tests were two-sided tests, and p<0.05 was considered significant.

## RESULT
### Characteristics of the included population
In total, 1764 classes from 81 primary schools and 88 middle schools (with 73 992 students) were randomly selected. Finally, 72 017 individuals completed the CBCL screening in stage 1, but 178 were excluded because their CBCLs were completed by the teachers or themselves rather than by the caregivers. 17 524 individuals (14 653 high-risk individuals and 2871 randomly selected non-high-risk individuals) underwent stage 2 assessment. The mean age of the included population was 11.56±5.12 years, and 66.2% lived in urban areas. The sex ratio of boys/girls was 0.99:1 (36 893/37 099). 18% of the total samples were diagnosed with at least one mental disorder. Among those who were not at high risk, 5.5% were diagnosed with at least one mental disorder.

### Point prevalence of anxiety disorder
The point prevalence of different subtypes of anxiety disorder and their 95% CIs are shown in table 1. The most prevalent anxiety disorder in children and adolescents was GAD, with an adjusted point prevalence of 1.3% (95% CI: 1.2 to 1.3). Followed by social phobia and separation anxiety disorder, the adjusted point prevalence was 0.8% (95% CI: 0.8 to 1.0) and 0.6% (95% CI: 0.5 to 1.0), respectively. The adjusted point prevalence of panic disorder was 0.2% (95% CI: 0.2 to 0.2). A total of 0.2% (95% CI: 0.1 to 0.2) were diagnosed with specific phobia, and 0.1% (95% CI: 0.1 to 0.1) were diagnosed with agoraphobia.

### The effect of age, gender and socioeconomic conditions on the prevalence of anxiety disorder
The participants were divided into children and adolescents group based on age. The age range of children was 6–12 years, and the age range of adolescents was 13–16 years. As listed in table 2, the distribution of anxiety disorder subtypes differed between different age groups. Adolescents had higher rates of panic disorder, GAD, agoraphobia and social phobia than that of children. Separation anxiety disorder and specific phobia were more common in children than in adolescents. The prevalence rate of separation anxiety was 0.90% (95% CI: 0.81

**Table 2** Differences in the prevalence of anxiety disorder in children and adolescents

| | Children (n=41 332) | | | Adolescents (n=30 775) | | | | |
|---|---|---|---|---|---|---|---|---|
| | Identified individuals | Point prevalence (%) | 95% CI | Identified individuals | Point prevalence (%) | 95% CI | $\chi^2$ | P |
| Panic disorder | 51 | 0.12 | 0.09 to 0.16 | 116 | 0.38 | 0.31 to 0.45 | 30.91 | **<0.001** |
| Generalised anxiety disorder | 431 | 1.04 | 0.95 to 1.15 | 475 | 1.54 | 1.41 to 1.69 | 8.76 | **0.003** |
| Specific phobia | 96 | 0.23 | 0.19 to 0.28 | 15 | 0.05 | 0.03 to 0.08 | 55.55 | **<0.001** |
| Agoraphobia without panic | 26 | 0.06 | 04 to 0.09 | 53 | 0.17 | 0.13 to 0.23 | 10.61 | **0.001** |
| Social phobia | 211 | 0.51 | 44 to 0.58 | 413 | 1.34 | 1.22 to 1.48 | 106.82 | **<0.001** |
| Separation anxiety disorder | 370 | 0.90 | 0.81 to 0.99 | 46 | 0.15 | 0.11 to 0.20 | 282.59 | **<0.001** |

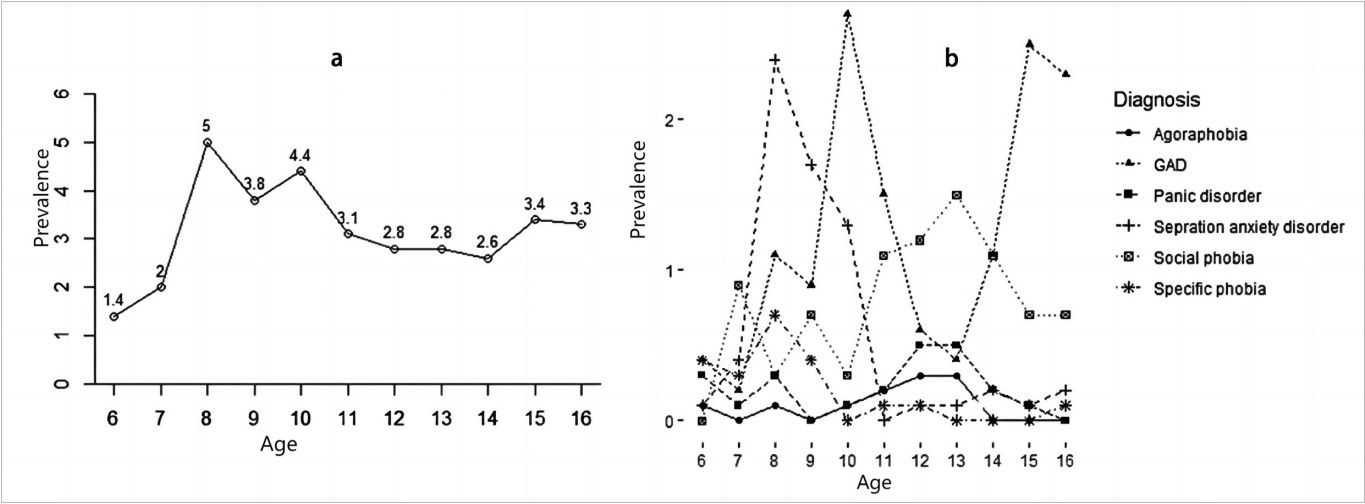

**Figure 2** Prevalence rate of anxiety disorder at different ages (GAD, generalised anxiety disorder).

to 0.99) in children and 0.15% (95% CI: 0.11 to 0.20) in adolescents ($\chi^2$=282.5, p<0.001). Specific phobia had a significantly higher rate in the child group (0.23% vs 0.05%, $\chi^2$=55.5, p<0.001) As shown in figure 2, age had a significant effect on the prevalence of different subtypes of anxiety disorder. The prevalence of anxiety disorder was relatively low before the age of 8 years. For different subtypes of anxiety disorder, the peak prevalence age for separation anxiety disorder was between 7 years and 10 years, and the prevalence was relatively low after 11 years old. The prevalence of social phobia rises sharply after the age of 10 years. The prevalence rate of anxiety disorder at different ages is demonstrated in online supplemental file 1.

The prevalence rates of panic disorder, GAD and agoraphobia were significantly different between gender groups. The prevalence of panic disorder in girls was one time higher than in boys. In the group of boys, the prevalence rate was 0.15% (95% CI: 0.11 to 0.19), while in the group of girls, it was 0.32% (95% CI: 0.26 to 0.38, $\chi^2$=14.6, p<0.001). Girls also had slightly higher rates of GAD and agoraphobia than that of boys. The prevalence

rate of agoraphobia was 0.08% (95% CI: 0.05 to 0.10) in the boy group and 0.14% (95% CI: 0.11 to 0.19) in the girl group ($\chi^2$=4.3, p=0.03). For GAD, the prevalence rate was 1.27% (95% CI: 1.16 to 1.39) in the girl group and 1.25% (95% CI: 1.13 to 1.36) in the boy group ($\chi^2$=4.71, p=0.02). There were no significant differences in the prevalence of specific phobia, social phobia and separation anxiety between boys and girls (for details, please see table 3).

Based on socioeconomic conditions, the five areas included in the study were divided into developed and underdeveloped areas. There was no significant difference in the prevalence of any anxiety disorder between these two groups. The results are listed in table 4.

### Comorbid conditions of anxiety disorder

A total of 70.3% (95% CI: 68.4 to 72.2) of the children were diagnosed with anxiety disorder comorbid with at least one other mental disorder. A total of 42.2% (95% CI: 40.2 to 44.3) had comorbidity with one other mental disorder, 27.5% (95% CI: 25.7 to 29.4) were comorbid with two other mental disorders and 0.6% (95% CI: 0.3

**Table 3** Differences in the prevalence of anxiety disorder between boys and girls

| | Boy (n=35 953) | | | Girl (n=36 154) | | | | |
|---|---|---|---|---|---|---|---|---|
| | Identified individuals | Point prevalence (%) | 95% CI | Identified individuals | Point prevalence (%) | 95% CI | $\chi^2$ | *P* |
| Panic disorder | 53 | 0.15 | 0.11 to 0.19 | 114 | 0.32 | 0.26 to 0.38 | 14.61 | **<0.001** |
| Generalised anxiety disorder | 448 | 1.25 | 1.13 to 1.36 | 458 | 1.27 | 1.16 to 1.39 | 4.71 | **0.02** |
| Specific phobia | 54 | 0.15 | 0.11 to 0.20 | 57 | 0.16 | 0.12 to 0.20 | 0.18 | 0.67 |
| Agoraphobia without panic | 27 | 0.08 | 0.05 to 0.10 | 52 | 0.14 | 0.11 to 0.19 | 4.30 | **0.03** |
| Social phobia | 290 | 0.81 | 0.72 to 0.90 | 334 | 0.92 | 0.83 to 1.02 | 0.01 | 0.93 |
| Separation anxiety disorder | 201 | 0.56 | 0.48 to 0.64 | 215 | 0.59 | 0.52 to 0.68 | 0.78 | 0.38 |

**Table 4** Differences in the prevalence of anxiety disorder between developed and less developed areas

| | Developed areas (n=10 215) | | | Less developed areas (n=61 892) | | | | |
|---|---|---|---|---|---|---|---|---|
| | Identified individuals | Point prevalence (%) | 95% CI | Identified individuals | Point prevalence (%) | 95% CI | $\chi^2$ | *P* |
| Panic disorder | 27 | 0.26 | 0.167 to 0.37 | 140 | 0.23 | 0.19 to 0.26 | 0.27 | 0.60 |
| Generalised anxiety disorder | 116 | 1.14 | 0.91 to 1.32 | 475 | 1.28 | 0.68 to 0.82 | 3.25 | 0.07 |
| Specific phobia | 19 | 0.19 | 0.11 to 0. 28 | 92 | 0.15 | 0.12 to 0.18 | 0.44 | 0.50 |
| Agoraphobia without panic | 12 | 0.12 | 0.06 to 0.20 | 67 | 0.11 | 0.08 to 0.13 | $1.9 \times 10^{-4}$ | 0.99 |
| Social phobia | 93 | 0.91 | 0.71 to 1.08 | 531 | 0.86 | 0.77 to 0.91 | 0.07 | 0.79 |
| Separation anxiety disorder | 67 | 0.66 | 0.49 to 0.81 | 349 | 0.56 | 0.49 to 0.62 | 0.90 | 0.34 |

to 1.0) were comorbid with three other mental disorders. The co-occurrence of anxiety disorder and ADHD, CD and oppositional defiant disorder (ODD) was very common. The most common comorbidity of GAD, specific phobia and separation anxiety disorder was ODD, with comorbid rates of 26.9%, 74.7% and 49%, respectively. The most common comorbidity for panic disorder was CD (24%). The ADHD-inattention subtype ranks as the most prevalent comorbidity for agoraphobia. Tic disorder was also common in anxiety disorder, and the comorbid rate of provisional tic disorder was 5.2% for GAD, 7.2% for panic disorder and 10.8% for specific phobia. Tourette's syndrome and chronic vocal tic disorder were also common in agoraphobia. The concurrent rates were 10.1% and 6.3%, respectively. In our survey, there were no comorbidities of anxiety disorder and eating disorder, adjustment disorder, psychotic disorders or pervasive developmental disorder (for details, please refer to figure 3).

## DISCUSSION

This first national survey carried out in China showed that GAD was the most common anxiety disorder in school children and adolescents, with a prevalence rate of 1.3%, followed by social phobia (0.9%), separation anxiety disorder (0.6%), panic disorder (0.2%), specific phobia (0.2%) and agoraphobia (0.1%). The distribution of anxiety disorder was different among different age and gender groups. Children had a higher prevalence rate of

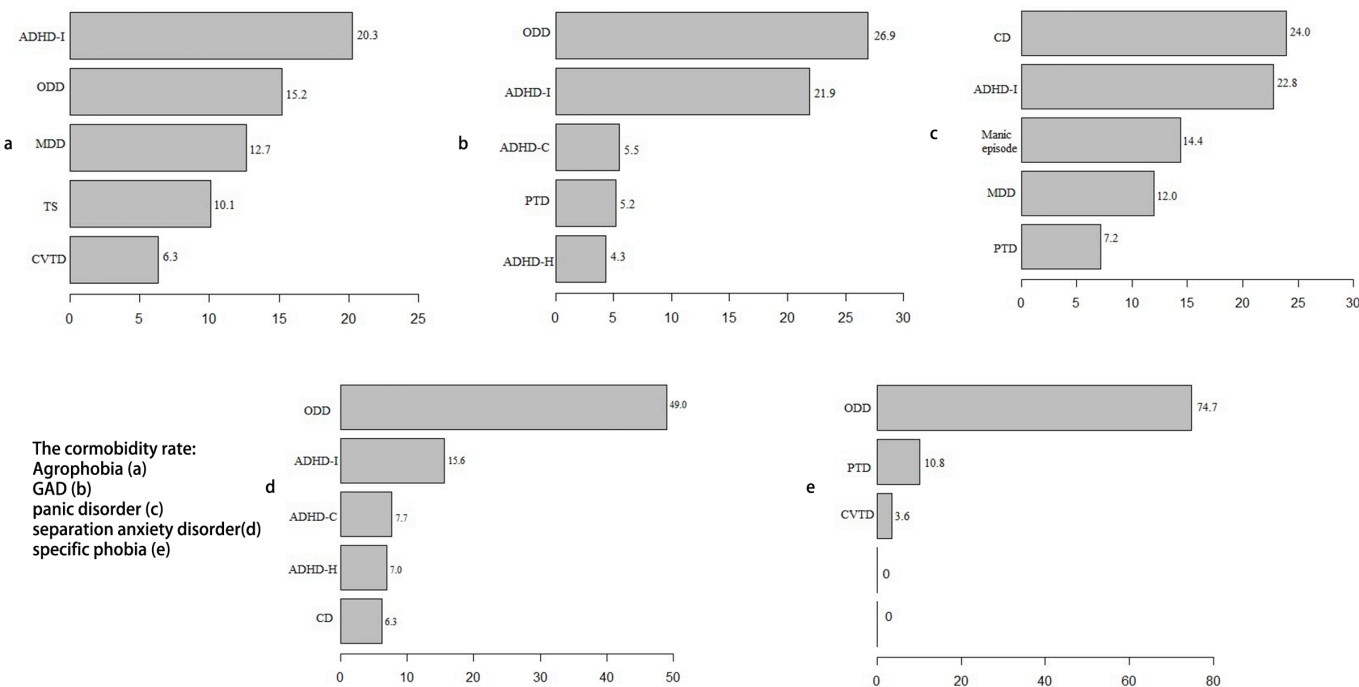

**Figure 3** Comorbid conditions of anxiety disorder. ADHD, attention-deficit/hyperactivity disorder; CD, conduct disorder; CVTD, chronic vocal tic disorder; GAD, generalised anxiety disorder; MDD, major depressive disorder; ODD, oppositional defiant disorder; TPTD, provisional tic disorder; Tourette's syndrome.

separation anxiety disorder and specific phobia than that of adolescents. Girls were more likely to be affected by panic disorder and GAD than boys. As for comorbidities, attention-deficit and disruptive behaviour disorders, tic disorders and depressive disorders were also commonly observed in the paediatric anxiety population.

Previous studies have shown that the prevalence of anxiety disorders in the paediatric population varies from 2% to 31% in different countries.[8–11 16 17] The prevalence of any anxiety disorder in our study was 4.7%. Our result was compatible with two other surveys carried out in Hunan and Sichuan provinces in China.[9 16] The prevalence rate was lower than that in many other studies.[11 17 18] The main reason might be related to the diagnostic criteria. In our survey, the DSM-IV criteria were used for the final diagnosis, which required functional impairment when making the diagnosis. A previous study showed that the prevalence of psychiatric disorders decreased sharply when the impairment criteria were applied.[18] In addition, stigma might be associated with low prevalence, as people in Asian countries are not used to reporting emotional problems.[27] Children and adolescents with self-stigma might not report their anxious experience.[28] As anxiety is an internal symptom, it might be ignored by caregivers.

Among the subtypes of anxiety disorder, GAD was the most prevalent anxiety disorder in our study (1.3%), which was consistent with two studies carried out in Hunan and Sichuan provinces, China. By investigating 17 071 primary and middle school students in Central Hunan province, Shen et al reported that the 12-month prevalence of GAD was 1.77%, followed by 0.74% for a specific phobia, 0.07% for social phobia and 0.02% for social anxiety disorder.[9] Roberts et al reported that agoraphobia was the most common anxiety disorder in their sample.[29] Several other studies reported that specific anxiety disorders had a much higher prevalence rate than other anxiety disorders.[11 17] However, the prevalence of panic disorder was relatively low (0.2%) in our study and other studies.[11 16] Notably, panic attacks are very common in the paediatric population. In the survey by Essau et al, 18% of adolescents reported experiencing panic attacks. However, only 0.5% met the diagnosis of panic disorder.[3]

Both age and gender had effects on the prevalence of any anxiety disorder. Our results showed that girls were more likely to be affected by GAD and panic disorder. Gender differences have been reported in many previous studies.[8 9 16] Wagner et al reported that the prevalence rate of anxiety disorder was one time higher in girls than in boys (19.53% and 9.52%, respectively).[10] However, some other studies found no gender difference.[1 2] The distribution of anxiety disorders was also different in different age groups. In general, adolescents had a higher rate of anxiety disorder, except for separation anxiety disorder. This was consistent with previous findings that anxiety disorder was less prevalent in younger children.[11 30] In our study, the peak age for separation anxiety disorder was between 7 years and 10 years. The finding was similar

to that of Mohammadi et al.[1] This might be related to the fact that at this time, children separated from their parents and started attending school. Most previous studies have shown that young children have a higher prevalence rate of separation anxiety disorder.[1 2] In contrast, the prevalence of social anxiety disorder increases with age.[30] Our results found no difference between developed areas and less developed areas. The effect of socioeconomic factors on the prevalence of anxiety disorder is not clear. A cross-nation survey indicated that socioeconomic conditions might be associated with the differences in the prevalence of anxiety subtypes, and high-income countries had higher prevalence rates of social anxiety disorder than low-income and lower-middle-income countries.[31] Many other studies failed to find differences between different socioeconomic status groups.[10 16]

In our study, 79.8% of children with anxiety disorder had at least one comorbidity. Previous studies indicated that the comorbid rate of anxiety disorder was high.[32] The comorbidity of other subtypes of anxiety disorder, depressive disorder, ADHD and disruptive, impulse control and CDs was common.[2 10 30] Our results showed that the most prevalent comorbidities of anxiety disorder were disruptive, impulse control and CDs. The most common comorbidity for GAD, specific phobia and separation anxiety disorder was ODD. CD ranked as the most common comorbidity for panic disorder. The high rate of comorbidity might be due to the following reasons. First, the symptoms of anxiety disorder and ODD and CD were highly associated.[33] Anger or irritability symptoms in the ODD lead to anxiety.[34] In addition, the functional impairment caused by ODD increases the risk of anxiety.[35] The concurrence of anxiety disorder and disruptive, impulse control and CDs would cause more severe functional impairment for the individual.[36] Comorbidity with ADHD was also very common in our study. The comorbidity of anxiety disorder and ADHD could be explained by the attentional control theory.[37] Individuals in an anxious state have increased attention to threat-related stimuli and impaired attentional inhibition and shifting skills.[38] Executive attention was impaired in children with anxiety disorders.[39] A neuroimaging study also confirmed deficits in prefrontal attentional control mechanisms in anxiety patients.[40]

The study had several limitations. First, the participants were enrolled in schools. Children who did not attend school were not included, and some of them may not attend school because of anxiety. This would undermine the representativeness of our sample. Second, the MINI-KID and the DSM-IV criteria were used for the diagnosis. However, the validity of the MINI-KID for the diagnosis of affective disorders, such as dysthymia, was not as good.[25] This may make the comorbidity results less comprehensive. The DSM-IV rather than DSM-5 diagnostic criteria was used in this study as it was started in October 2014, and the Chinese version of DSM-5 was published in July 2014. There were not enough time and resources to train clinicians to perform the DSM-5

assessment. The classification of anxiety disorder subtypes had changed a lot in DSM-5 criteria. For example, selective mutism was categorised as an anxiety disorder in DSM-5 but not in DSM-IV. This would limit the comparability of the results with recent studies based on DSM-5. Third, it should be noted that all diagnoses were made retrospectively, and recall bias could be a confounding factor for the results. Although according to the DSM-IV diagnostic criteria, functional impairment was considered while making the final diagnosis, there still lacks explicit measurement for functional impairment using psychological scales such as clinical global impression; this is another limitation of the study.

This study was a two-stage, national epidemiological survey using a stratified cluster sampling method. The participants represented school children and adolescents from different socioeconomic backgrounds. The results demonstrated that anxiety disorder was common among school students. Girls were more likely to be affected by anxiety disorder than boys. The distribution of anxiety changed with age. Caregivers and teachers should pay more attention to the risk emotions of children. Our results also revealed that ADHD, ODD and CD often co-occurred with anxiety disorder. This implies that for children with existing external symptoms such as aggressive symptoms and hyperactivity, the assessment of internal symptoms should not be neglected. A comprehensive assessment should be carried out in case the anxiety disorder is masked by external symptoms.

**Author affiliations**
[1]Department of Psychiatry, Beijing Children's Hospital, Capital Medical University, National Center for Children's Health, Beijing, China
[2]Beijing Language and Culture University, Beijing, China
[3]Chinese Academy of Sciences, Beijing, China
[4]Department of Pediatrics, Beijing Anding Hospital, Capital Medical University, Beijing, China
[5]Big Data Center, Beijing Children's Hospital, Capital Medical University, National Center for Children's Health, Beijing, China
[6]Beijing Children's Hospital, Capital Medical University, National Center for Children's Health, Beijing, China

**Contributors** YZ and YC were responsible for the design and carry-out of the survey. FW contributed to the data analysis and writing of the manuscript. HX and YL collaborated in the writing of the manuscript. HY polished the language. RW, FL and YL participated in collecting data and digitalising the data.

**Funding** This work was supported by the National Natural Science Foundation of China (under Grant No. 82001445, No. 82171538), the Natural Science Foundation of Beijing Municipality (under Grant No. 7212035), Beijing Hospitals Authority Youth Programme (Grant No. QML20211203) and Beijing Chronic Disease Prevention and Health Education Research Programme (under Grant No. BJMB0012022028019).

**Competing interests** None declared.

**Patient and public involvement** Patients and/or the public were not involved in the design, or conduct, or reporting, or dissemination plans of this research.

**Patient consent for publication** Not applicable.

**Ethics approval** This study involves human participants. The study was conducted in accordance with the Declaration of Helsinki, and the protocol was approved by the Ethics Committee of the Anding Hospital, Capital Medical University (project identification code: 2012BAI01B02). Participants gave informed consent to participate in the study before taking part.

**Provenance and peer review** Not commissioned; externally peer reviewed.

**Data availability statement** Data are available upon reasonable request.

**ORCID iD**
Yonghua Cui http://orcid.org/0000-0002-8244-5884

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
