## [Reviewer comments · BMJ Paediatrics Open]

ARTICLE DETAILS

TITLE (PROVISIONAL)	Prevalence and comorbidity of anxiety disorder in school-attending children and adolescents aged 6-16 in China
AUTHORS	Wang, Fang Yang, Hanxue Li, Fenghua Zheng, Yi Xu, Hui Wang, Rui Li, Ying Cui, Yonghua

VERSION 1 – REVIEW

REVIEWER	Dr. Surendra Gupta Childrens Medical Center of Fresno, Pediatrics
REVIEW RETURNED	26-Jun-2023

GENERAL COMMENTS	The abstract provides an overview of a nationwide epidemiological survey conducted in China to determine the prevalence and characteristics of pediatric anxiety disorders. However, there are some areas where the study could be improved: Clearer research objectives: The abstract lacks a clear statement of the research objectives or hypotheses. It would be helpful to explicitly state the purpose of the study, such as investigating the prevalence, comorbidity rates, and distribution profiles of anxiety disorders among school students in China. Introduction section: The introduction provides a general background on anxiety disorders but does not clearly state the gap in knowledge or the need for the study. It would be beneficial to clearly highlight the significance of conducting a national epidemiological survey in China and emphasize the lack of previous nationwide data on pediatric anxiety disorders. Methods section: The methods section should provide more details about the sampling strategy, including the specific criteria for selecting the provinces and the rationale for considering them as representative of developed and developing areas in China. Conclusion: The abstract briefly mentions the findings but does not explicitly state the main conclusions of the study. It would be helpful to summarize the key findings and their implications for clinical practice, policy, and future research. Lack of discussion on limitations: The study provided a detailed discussion of the findings but did not explicitly discuss the
---

	limitations of the review itself. Addressing the limitations would have enhanced the transparency and credibility of the study. Here are some limitations of the study based on the provided information: Sampling bias: The study used a multistage cluster sampling method to select participants from five provinces in China. However, the representativeness of the sample is unclear. The abstract does not provide information on how the provinces were selected or whether they are representative of the entire population. This could introduce bias and limit the generalizability of the findings. Screening tool limitations: The study used the Child Behavior Checklist (CBCL) as a screening tool to identify high-risk individuals. While the Chinese version of the CBCL has been shown to have good reliability and validity, the abstract does not provide information on the sensitivity and specificity of the CBCL for detecting anxiety disorders specifically. Using a screening tool with limited accuracy could lead to misclassification of individuals and affect the prevalence estimates. Diagnostic criteria: The study used the DSM-IV criteria for diagnosis, which is an older version of the diagnostic manual. The use of outdated diagnostic criteria may affect the accuracy and comparability of the results with more recent studies using updated criteria (e.g., DSM-5). The abstract does not mention whether standardized diagnostic interviews were conducted by trained clinicians or if interrater reliability was assessed. Inaccurate or inconsistent application of diagnostic criteria could affect the reliability of the prevalence estimates. Lack of functional impairment assessment: The abstract does not mention whether functional impairment was considered when making the diagnosis of anxiety disorders. Functional impairment is an important criterion for diagnosing anxiety disorders and could impact the prevalence estimates. Its omission raises questions about the accuracy of the diagnosis and the severity of the reported cases. Limited information on comorbidity: The abstract mentions that comorbidity rates between anxiety disorders and other mental disorders were calculated, but it does not provide specific details about the comorbid conditions or the methodology used to assess comorbidity. Without this information, it is challenging to evaluate the impact of comorbidity on the prevalence estimates and the validity of the findings. Lack of socioeconomic factors: Although the study compares the prevalence rates of anxiety disorders across different age and gender groups, it does not provide detailed information on the socioeconomic factors assessed or their potential influence on the prevalence estimates. This limits the understanding of how socioeconomic factors may contribute to the observed rates of anxiety disorders. Lack of external validation: The study did not compare its findings with other independent sources of data or replicate the study using
--	--

	different methodologies. External validation and replication are essential for establishing the reliability and generalizability of the findings. Lack of Intervention or Treatment Information: The study focused on the prevalence and characteristics of anxiety disorders but did not provide information on interventions or treatment outcomes. Understanding the availability and effectiveness of interventions for anxiety disorders in the studied population would be valuable for informing healthcare policies and practices. Lack of follow-up data: The study focused on the point prevalence of anxiety disorders and did not provide information on the longitudinal course or outcomes of these disorders. Longitudinal data would provide a better understanding of the stability, remission, or recurrence of anxiety disorders over time. Lack of external validation: The study did not compare its findings with other independent sources of data or replicate the study using different methodologies. External validation and replication are essential for establishing the reliability and generalizability of the findings.
--	--

REVIEWER	Dr. Sarah Jane Nevitt University of York, Centre for Reviews and Dissemination
REVIEW RETURNED	04-Jul-2023

GENERAL COMMENTS	I have conducted a statistical review of the manuscript "Prevalence and comorbidity of anxiety disorder in school-attending children and adolescents aged 6-16 in China" The authors present an analysis of their epidemiological survey of mental disorders among school students aged 6-16 years in China estimating prevalence rates of anxiety disorders. Overall, the survey seems to be well designed (the design has been previously published) and the current analysis well conducted and presented. I have a few points of clarification:  1. Introduction: "Children with anxiety disorder were more likely to have depression disorder..." Other statements in this section are made in the present tense "anxiety disorders have..." "anxiety disorders are..." so I suggest "were" should be changed to "are" in the highlighted sentence for consistency 2. Introduction "The prevalence of any anxiety disorder among adolescents in Hong Kong was estimated to be 30.2%[20]." Was an age range provided to define adolescents in this study? 3. Please define the abbreviation DSM-IV at the first use 4. Methods (2.1): "Detailed information, including the sampling design, sample size calculation, and socio-demographic characteristics of the sampling areas included, has been described in previous studies (see Supplementary Materials)."
---

	The only Supplementary document available is list of previous epidemiological studies of anxiety disorder in paediatric population. No details of sampling design, sample size calculation, and socio-demographic characteristics of the sampling areas included is available? Is a Supplementary document missing? 5. Methods (2.2): “According to the previous study of Liu et al., we chose a cut-off score of 35...” Should reference 31 be cited in this sentence? 6. Methods (2.3): “Considering the false-negative rate of CBCL screening,...” What is the false-negative rate? 7. Methods (2.3): I am confused regarding the timings of the interviews and the diagnosis in this study. Did all participants in Stage 2 assessment have both a MINI-KID and DSM-IV assessment? Figure 1 reads like the MINI-KID interviews were conducted (n=17,524), those who were negative according to the MINI-KID (n=4277) were then excluded and did not have a DSM-IV interview? If this was the process, what is the false negative rating of the MINI-KID? Section 2.3 implies that different psychiatrists performed the MINI-KID and DSM-IV interviews? And the psychiatrist performing the DSM-IV interview was responsible for the final diagnosis? The limitations of the study also refers to diagnoses being made retrospectively and the risk of recall bias. My understanding was that the survey and the diagnosis are part of the same study, and that the diagnoses reflected a 'current' disorder at the time of the study? Do the authors mean that the MINI-KID and DSM-IV interviews were conducted, and at a later date the diagnoses were made? If specific criteria (i.e. DSM-IV) are used and this information was recorded at the time of interview then recall bias should not be an issue. Please clarify the timings of interviews and diagnoses. 8. Methods (2.5): “The prevalence was adjusted based on the weights of their provincial region, prefectural division, county/district, school, and class.” I' don't understand what is meant by adjusted here. If the appropriate denominator is used to calculate prevalence for each region, division etc. then no adjustment should be required? Please clarify. 9. Methods (2.5): Please add a reference for Rao-Scott adjusted chi-square tests
--	--

10. Results (3.1) : “Finally, 71929 individuals completed the CBCL screening in stage 1...” This should be 72,107 according to Figure 1?

11. Comparisons of boys and girls are reported as ratios both for characteristics (Section 3.1) and when summarising the results for panic disorder and GAD (Section 4). I don't think that a relative measure which compares numbers of boys and girls / results for boys and girls without stating what the numbers or results are is meaningful in this context. It is also confusing when other numbers and results are presented as percentages. Please present the proportions of boys and girls in the population in Section 3.1 and the calculated prevalence results in Section 4

12. Figure 1: Please correct the number of individuals diagnosed with GAD in Figure 1, I think this should read 906 to match Table 1.

The totals in Table 1 of the different disorders diagnosed sum to 2,303 (this is before overlap if any individuals were diagnosed with more than one anxiety disorder), so were the remaining 10,727 represented in Figure 1 diagnosed with other types of mental disorder? (i.e. not anxiety disorders?)

Please clarify, some edits to Figure 1 may be required such as an additional cell in the bottom row to show 'Diagnosed with other non-anxiety mental disorder (n=)'

13. Table 2: What were the age ranges for 'children' and 'adolescents' in this study?

14. Figure 2: The footnotes to the figures seem to be missing so I'm not sure what Figure 2a is showing? The prevalence of any anxiety disorder?

Can this be added to Table 1 with the subtypes of anxiety disorder?

15. Table 4 - the numbers of people in developed areas and less developed areas add up to 73992 (i.e. included participants) while the totals in the other tables of children and adolescents / boys and girls add up to 72107 (i.e. participants who underwent CBCL screening). Is there a reason for this difference?

16. Results (3.4): Were any children diagnosed with more than one anxiety disorder?

17. References: Please check the references, for example the journal for reference 29, the previous study of this survey is missing.

When referring to journal articles, a consistent referencing style which includes authors, title and journal information should be used rather than a weblink to a PubMed page. Only websites should be cited in this way.

VERSION 1 – AUTHOR RESPONSE

Reviewer: 1

Dr. Surendra Gupta, Children's Medical Center of Fresno, Saint Agnes Medical Center

1. Clear research objectives: The abstract lacks a clear statement of the research objectives or hypotheses. It would be helpful to explicitly state the purpose of the study, such as investigating the prevalence, comorbidity rates, and distribution profiles of anxiety disorders among school students in China.

*Response: Thanks a lot for your suggestion. We have revised the abstract and clearly stated the purpose of this study. In the background, we stated the reported prevalence rate of anxiety disorder in the pediatric population varies widely between different counties. Currently, there is no national epidemiological survey of childhood anxiety disorder in China. This study aims to investigate the national prevalence of anxiety disorder, the distribution profiles of different subtypes, and its comorbidity rates among school students.

2. Introduction section: The introduction provides a general background on anxiety disorders but does not clearly state the gap in knowledge or the need for the study. It would be beneficial to clearly highlight the significance of conducting a national epidemiological survey in China and emphasize the lack of previous nationwide data on pediatric anxiety disorders.

*Response: Thank you for your comment. We have revised the introduction section to highlight the fact that there is no national epidemiological survey of childhood anxiety disorder in China currently. We also stated the significance of this study, to fill the gap in anxiety disorder prevalence in school-aged children and adolescents in China.

We stated at the beginning of the third paragraph of the introduction that at present, the national prevalence rate of anxiety disorders in the pediatric population in China is still not clear as there is no nationwide epidemiological survey. There are only a few studies conducted locally. And added summarized the results vary a lot and cannot represent the overall situation in China. It is of great importance to estimate the national prevalence rate of pediatric anxiety disorder as it will offer data support for government officials and mental health service providers.

3. Methods section: The methods section should provide more details about the sampling strategy, including the specific criteria for selecting the provinces and the rationale for considering them as representative of developed and developing areas in China.

*Response: Thanks for your comment. We had expanded the methods. We described the sampling method and the reason why we chose these five provinces in detail.

4. Conclusion: The abstract briefly mentions the findings but does not explicitly state the main conclusions of the study. It would be helpful to summarize the key findings and their implications for clinical practice, policy, and future research.

*Response: Thank you for your suggestion. We revised the conclusion of the abstract and stated that anxiety disorder was prevalent among school students in China and comorbidity with attention deficit and disruptive disorder was very common. The data implied screening for anxiety disorder is needed in school settings. Policies should be adapted to provide psychological services to children and adolescents. A comprehensive assessment is recommended in clinical practice in the conclusion.

Lack of discussion on limitations: The study provided a detailed discussion of the findings but did not explicitly discuss the limitations of the review itself. Addressing the limitations would have enhanced the transparency and credibility of the study.

Here are some limitations of the study based on the provided information:

5. Sampling bias: The study used a multistage cluster sampling method to select participants from five provinces in China. However, the representativeness of the sample is unclear. The abstract does not provide information on how the provinces were selected or whether they are representative of the entire population. This could introduce bias and limit the generalizability of the findings.

*Response: Thank you for your comment. We added the sampling method in detail in the method part of the main text. Due to the word limitation of the abstract, we could not provide this information in the abstract. However, we added this point in the limitation in the section of Discussion.

6. Screening tool limitations: The study used the Child Behavior Checklist (CBCL) as a screening tool to identify high-risk individuals. While the Chinese version of the CBCL has been shown to have good reliability and validity, the abstract does not provide information on the sensitivity and specificity of the CBCL for detecting anxiety disorders specifically. Using a screening tool with limited accuracy could lead to misclassification of individuals and affect the prevalence estimates.

*Response: Thank you for your comment. A previous meta-analysis had shown the CBCL internalizing syndrome subscale had good discriminant validity for anxiety disorder. The weighted mean effect size was 1.55 (95% CI: 1.10-2.01). we had added this in the main text.

7. Diagnostic criteria: The study used the DSM-IV criteria for diagnosis, which is an older version of the diagnostic manual. The use of outdated diagnostic criteria may affect the accuracy and comparability of the results with more recent studies using updated criteria (e.g., DSM-5).

*Response: Thanks for your comment. This study started in October 2014 and the Chinese version of DSM-5 was published in July 2014, there was not enough time and resources to train clinicians to perform the DSM-5 assessment. This is truly a limitation, and we added this in the discussion part.

8. The abstract does not mention whether standardized diagnostic interviews were conducted by trained clinicians or if interrater reliability was assessed. Inaccurate or inconsistent application of diagnostic criteria could affect the reliability of the prevalence estimates.

*Response: Thank you for your suggestion. 250 psychiatrists were trained to perform the MINI-KID assessment, and the inter-rater reliability was also evaluated, we have added this information in the main text-method part. Due to the word limitation of the abstract, we did not add this information to the abstract.

9. Lack of functional impairment assessment: The abstract does not mention whether functional impairment was considered when making the diagnosis of anxiety disorders. Functional impairment is an important criterion for diagnosing anxiety disorders and could impact the prevalence estimates. Its omission raises questions about the accuracy of the diagnosis and the severity of the reported cases.

*Response: Thank you for your suggestion. We are sorry we did not state it clearly. The final diagnosis was made based on the DSM-IV, and functional impairment was one of the diagnostic criteria. We have added this in the main text. As it is an epidemiological study, we did not have explicit measurements for functional impairment using psychological scales such as clinical global impression (CGI). We added it as a limitation of the study.

10. Limited information on comorbidity: The abstract mentions that comorbidity rates between anxiety disorders and other mental disorders were calculated, but it does not provide specific details about the comorbid conditions, or the methodology used to assess comorbidity. Without this information, it is challenging to evaluate the impact of comorbidity on the prevalence estimates and the validity of the findings.

*Response: Thanks for your comment. Due to the word limitation of the abstract, we did not list the comorbid rate of anxiety disorder explicitly. However, the details were listed in the results part (3.4) in the main text.

11. Lack of socioeconomic factors: Although the study compares the prevalence rates of anxiety disorders across different age and gender groups, it does not provide detailed information on the socioeconomic factors assessed or their potential influence on the prevalence estimates. This limits the understanding of how socioeconomic factors may contribute to the observed rates of anxiety disorders.

*Response: Thanks for your comment. We compared the prevalence of anxiety disorder between developed and developing Ares, and the results were listed in Table 4. The socioeconomic conditions of the included provinces were described in detail in the method part.

11. Lack of external validation: The study did not compare its findings with other independent sources of data or replicate the study using different methodologies. External validation and replication are essential for establishing the reliability and generalizability of the findings.

*Response: Thank you for your comment. In the discussion, we compared our results with previous epidemiological studies conducted in China and other countries and discussed about the possible reason for the difference.

Previous studies have shown that the prevalence of anxiety disorders in the pediatric population varies from 2% to 31% in different countries[reference 15–18,21,23]. The prevalence rate was lower than that in many other studies. The main reason might be related to the diagnostic criteria. In our survey, the DSM-IV criteria were used for the final diagnosis, which required functional impairment when making the diagnosis. A previous study showed that the prevalence of psychiatric disorders decreased sharply when the impairment criteria were applied[20,24]. In addition, stigma might be associated with low prevalence, as people in Asian countries are not used to reporting emotional problems[37,38]. Children and adolescents with self-stigma might not report their anxious experiences [39]. As anxiety is an internal symptom, it might be ignored by caregivers.

12. Lack of Intervention or Treatment Information: The study focused on the prevalence and characteristics of anxiety disorders but did not provide information on interventions or treatment outcomes. Understanding the availability and effectiveness of interventions for anxiety disorders in the studied population would be valuable for informing healthcare policies and practices.

*Response: Thank you for your suggestion. This study was epidemiological, the aim was to estimate the national prevalence rate of anxiety disorder in the pediatric population in China, as such data is lacking currently. The design of this study did not include intervention for anxiety disorder. We hope we can explore the effectiveness of interventions for anxiety disorders in the studied population in the future. We added this in the discussion as a perspective for future study.

13. Lack of follow-up data: The study focused on the point prevalence of anxiety disorders and did not provide information on the longitudinal course or outcomes of these disorders. Longitudinal data would provide a better understanding of the stability, remission, or recurrence of anxiety disorders over time.

*Response: Thank you for your suggestion. The aim was to estimate the national prevalence rate of anxiety disorder in the pediatric population in China, as such data is lacking currently. It was a cross-sectional study and estimated the point prevalence. we will include the follow-up in our future study as we can provide a longitudinal picture of the stability, remission, or recurrence of anxiety disorders over time. We added this in the discussion as a perspective for the future study.

14. Lack of external validation: The study did not compare its findings with other independent sources of data or replicate the study using different methodologies. External validation and replication are essential for establishing the reliability and generalizability of the findings.

*Response: Thank you for your comment. In the discussion, we compared our results with previous epidemiological studies conducted in China and other countries and discussed about the possible reason for the difference.

Reviewer: 2

Dr. Sarah Nevitt, University of Liverpool

Comments to the Author

I have conducted a statistical review of the manuscript "Prevalence and comorbidity of anxiety disorder in school-attending children and adolescents aged 6-16 in China"

The authors present an analysis of their epidemiological survey of mental disorders among school students aged 6-16 years in China estimating prevalence rates of anxiety disorders.

Overall, the survey seems to be well designed (the design has been previously published) and the current analysis is well conducted and presented. I have a few points of clarification:

1. Introduction: "Children with anxiety disorder were more likely to have depression disorder..."

Other statements in this section are made in the present tense "anxiety disorders have..." "anxiety disorders are..." so I suggest should be changed to "are" in the highlighted sentence for consistency

*Response: Thank you for your comment. We have changed the "were" in the sentence "Children with anxiety disorder were more likely to have depression disorder, suicidal behavior, and other psychiatric disorders in adulthood" into "are".

2. Introduction "The prevalence of any anxiety disorder among adolescents in Hong Kong was estimated to be 30.2%[20]." Was an age range provided to define adolescents in this study?

Response: Thanks for your comment. The age range of the population included in the study of Leung et.al was not clearly stated. There were Grade 7 to 9 high school students, and the mean age was 13.8 years (SD = 1.2). We have added the mean age information in the main text.

3. Please define the abbreviation DSM-IV at the first use

Response: Thanks for your suggestion. The first use of DSM-IV was in the last paragraph of the introduction section. We have added the definition of this abbreviation.

4. Methods (2.1): "Detailed information, including the sampling design, sample size calculation, and socio-demographic characteristics of the sampling areas included, has been described in previous studies (see Supplementary Materials)."

The only Supplementary document available is a list of previous epidemiological studies of anxiety disorder in the pediatric population. No details of sampling design, sample size calculation, and socio-demographic characteristics of the sampling areas included are available. Is a Supplementary document missing?

*Response: Thank you for your comment. We expanded the method and described the sampling method, size calculation, and socioeconomic conditions of the included provinces in detail.

This is a two-stage survey carried out in primary and high schools in Beijing, Liaoning, Jiangsu, Sichuan, and Hunan Provinces. The age range of school students is from 6 to 17 years. Due to the consideration of the College Entrance Examination, we decided not to include students in the final year of secondary school. So, the age range of students included in this survey was 6 to 16 years. The five provincial administrative regions were chosen by taking the geographical partition, economic development, and rural/urban factors into consideration. In the official document, (National Bureau of Statistics of China, 2011), China has divided the country into northeastern, eastern, middle, and western regions geographically. Liaoning was chosen as the representative of the northeast area and Jiangsu as the representative of the eastern area, Sichuan as the representative of the western area, Hunan as the representative of the southern area. Beijing was picked as being representative of a Developed Urban Area, which had a gross domestic product per capita of over 12,736 US dollars, and a population larger than 10 million. Then we randomly selected 2–4 prefecture divisions in these five areas. Fifteen prefecture divisions in total were selected finally. Next, in each prefectural division, simple sampling without replacement was used to select schools one by one (with a predetermined ratio of primary schools to middle schools as 1:1). Then, we randomly selected 2–5 classes in each grade of every school. The sample size was calculated using the formula $n = (z_{(1-\alpha/2)}^2 (1-p)) / (p\epsilon^2)$

). We chose the prevalence of Tourette disorder (0.30%) to estimate the sample size as it was reported to be the lowest among children and adolescents aged 6–17[30]. with a confidence coefficient of 95% (Z) and relative error (ϵ) of 15%, 56742 participants would reach a statistic power of 1. We assumed 20% would be lost during the follow-up and decided to recruit 73992 participants. Based on this, we randomly chose 1764 classes from 169 schools (81 primary schools and 88 high schools)

5. Methods (2.2): “According to the previous study of Liu et al., we chose a cut-off score of 35...” Should reference 31 be cited in this sentence?

*Response: Thank you for your comment. Yes, reference 31 is cited in this sentence, it was cited at the end of the sentence.

6. Methods (2.3): “Considering the false-negative rate of CBCL screening,...” What is the false-negative rate?

Response: Thanks for your suggestion. Yes, we randomly chose 5% of the participants whose CBCL showed negative results to assess the false negative rate of CBCL. At last, 2,871 nonhigh-risk individuals were selected and underwent the stage 2 interview, our results showed 158 of them were diagnosed as having at least 1 mental disorder, so the false-negative rate was 5.5%.

7. Methods (2.3): I am confused regarding the timings of the interviews and the diagnosis in this study. Did all participants in the Stage 2 assessment have both a MINI-KID and DSM-IV assessment?

Response: Thanks for your comment. Yes, all the participants in Stage 2 underwent two interviews, the MINI-KID, and the DSM-IV interview. The first one was the MINI-KID interview, and the rating results of MINI-KID were given to the clinicians who conducted DSM-IV interviews as references. The final diagnoses were made by the psychiatric clinicians conducting DSM-IV interviews.

8. Figure 1 reads like the MINI-KID interviews were conducted (n=17,524), those who were negative according to the MINI-KID (n=4277) were then excluded and did not have a DSM-IV interview. If this was the process, what would be the false negative rating of the MINI-KID?

Response: Thanks for your suggestion. Yes, in stage 2, 4277 participants did not undergo DSM-IV interviews as their results of MINI-KID interviews were negative. Participants with CBCL score higher than 35 would be included in the stage 2 interview. In stage 2, the MINI-KID interviews were first carried out as a screening interview, and the diagnosis made by the MINI-KID interviews would then be further checked by psychiatrists according to DSM-IV. Those who showed negative MINI-KID results, which means they were not affected by any mental disorder according to MINI-KID interviews would not undergo DSM-IV interviews.

9. Section 2.3 implies that different psychiatrists performed the MINI-KID and DSM-IV interviews. And the psychiatrist performing the DSM-IV interview responsible for the final diagnosis?

Response: Thanks for your comment. Yes, the final diagnoses were made by the psychiatric clinicians conducting DSM-IV interviews. This MINI-KID interview and DSM-IV interview are not the same interviewers. The psychiatrists performed the MINI-KID and DSM-IV interviews from different groups.

10. The limitations of the study also refer to diagnoses being made retrospectively and the risk of recall bias. My understanding was that the survey and the diagnosis were part of the same study and that the diagnoses reflected a 'current' disorder at the time of the study. Do the authors mean that the MINI-KID and DSM-IV interviews were conducted, and at a later date the diagnoses were made? If specific criteria (i.e. DSM-IV) are used and this information was recorded at the time of the interview then recall bias should not be an issue. Please clarify the timings of interviews and diagnoses.

Response: Thank you for your comment. The diagnoses were made immediately after the DSM-IV interviews were completed, and it was based on the status of the participants. But, according to DSM-IV, the time periods for the diagnosis of different mental disorders were different. For example, for the diagnosis of social phobia and separation anxiety disorder, the symptoms need to last for at least 4 weeks, and the time for generalized anxiety disorder was 6 months. For some disorders such as conduct disorder, the time was as long as 12 months. The participants and their parents had to recall the behavior and might have the risk of recall bias.

11. Methods (2.5): "The prevalence was adjusted based on the weights of their provincial region, prefectural division, county/district, school, and class." I don't understand what is meant by adjusted here. If the appropriate denominator is used to calculate prevalence for each region, division, etc. then no adjustment should be required. Please clarify. 12. Methods (2.5): Please add a reference for Rao-Scott adjusted chi-square tests

Response: Thanks a lot for your comment. The adjustment was based on the product of sampling weights and poststratification weights. 1 was the product of sampling weights of their provincial region, prefectural division, county/district, school, and class. In Stage 2, the weights of randomly picked participants with negative CBCL screening results were multiplied by the reciprocal of their sampling probabilities. Poststratification weight was calculated by utilizing the demographic characteristics of participants: (a) age group (6–11 years and 12–16 years), (b) sex, and (c) location of residence (urban vs. rural). Nonresponses adjustment was included in the poststratification process. Individuals who refused to participate or did not have their primary caregivers finish the CBCL were treated as non-respondents. We used the reciprocal of the response rate in the corresponding demographic sub-group of each participant as their nonresponse weights.

13. Results (3.1): "Finally, 71929 individuals completed the CBCL screening in stage 1..." This should be 72,107 according to Figure 1.

Response: Thanks for your suggestion. Yes, 72107 in Figure 1 was the number of participants that underwent CBCL screening, but 178 individuals were excluded because their CBCL were not completed by their caregivers, so 71929 individuals completed the CBCL screening in stage 1. We are sorry for the misunderstanding it caused and revised the Figure 1. we also revised the description in the Results (3.1)

14. Comparisons of boys and girls are reported as ratios both for characteristics (Section 3.1) and when summarising the results for panic disorder and GAD (Section 4). I don't think that a relative measure that compares numbers of boys and girls/results for boys and girls without stating what the numbers or results are is meaningful in this context. It is also confusing when other numbers and results are presented as percentages. Please present the proportions of boys and girls in the population in Section 3.1 and the calculated prevalence results in Section 4

Response: Thanks for your suggestion. We are not sure whether we understand you correctly. We have added the number of boys and girls in the population in section 3.1 when describing the characteristics of the included population. We compared the prevalence difference between boys and girls and stated the results in the second paragraph of section 3.3 and Table 4, where the point prevalence was listed.

12. Figure 1: Please correct the number of individuals diagnosed with GAD in Figure 1, I think this should read 906 to match Table 1.

Response: Thanks for your suggestion. We are sorry for our mistake, and we have corrected Figure 1.

13. The totals in Table 1 of the different disorders diagnosed sum to 2,303 (this is before overlap if

any individuals were diagnosed with more than one anxiety disorder), so were the remaining 10,727 represented in Figure 1 diagnosed with other types of mental disorder? (i.e. not anxiety disorders?) Please clarify, some edits to Figure 1 may be required such as an additional cell in the bottom row to show 'Diagnosed with other non-anxiety mental disorder (n=)'

Response: Thank you for your suggestion. There is comorbidity with other mental disorders for the 2303 diagnosed with anxiety disorder. The remaining 10,727 were diagnosed with other mental disorders. We have added this in Figure 1.

14. Table 2: What were the age ranges for 'children' and 'adolescents' in this study?

Response: Thanks for your suggestion. The age range of children was 6 to 12 years old, and the age range of adolescents was 13 to 16, we added this information in the manuscript (the results part, 3.3).

15. Figure 2: The footnotes to the figures seem to be missing so I'm not sure what Figure 2a is showing. The prevalence of any anxiety disorder?

Can this be added to Table 1 with the subtypes of anxiety disorder?

Response: Thanks for your comment. Figure 2a shows the prevalence rate of anxiety disorder in total at different ages. It is impossible to add this in Table 1, but we added sTable 2 to illustrate the exact prevalence rate.

16. Table 4 - the numbers of people in developed areas and less developed areas add up to 73992 (i.e. included participants) while the totals in the other tables of children and adolescents/boys and girls add up to 72107 (i.e. participants who underwent CBCL screening). Is there a reason for this difference?

Response: Thanks a lot for your comment. We are sorry for our mistake. The numbers of people in developed areas and less developed areas in Table 4 were included participants, but the prevalence rate should be calculated based on the number who finished CBCL screening, we have revised the number in Table 4.

17. Results (3.4): Were any children diagnosed with more than one anxiety disorder?

Response: Thank you for your comment. No, in our data, there were no children diagnosed with more than one anxiety disorder.

18. References: Please check the references, for example, the journal for reference 29, the previous study of this survey is missing.

When referring to journal articles, a consistent referencing style that includes authors, title and journal information should be used rather than a weblink to a PubMed page. Only websites should be cited in this way.

Response: Thank you for your suggestion. We have added the previous survey of reference 29, and we have checked the references and made sure the style is consistent.

VERSION 2 – REVIEW

REVIEWER	Dr. Sarah Jane Nevitt University of York, Centre for Reviews and Dissemination
REVIEW RETURNED	22-Nov-2023
GENERAL COMMENTS	Thank you to the authors for their responses to my comments and revisions to their manuscript. I have a couple of minor follow-up comments related to my original comments.

	Original comment 11: Thank you for providing further information about the calculation of prevalence using adjusted weights in your response. Please add some further information to this manuscript about this approach, or if these methods have been previously published, please add a reference. This may be helpful to a reader wishing to understand the methods used here, perhaps wishing to conduct a similar study. References: Please check the references again. The Journal is missing for reference 26 and reference 7, the authors and journal are missing.
--	---

VERSION 2 – AUTHOR RESPONSE

1、 Thank you for providing further information about the calculation of prevalence using adjusted weights in your response. Please add some further information to this manuscript about this approach, or if these methods have been previously published, please add a reference. This may be helpful to a reader wishing to understand the methods used here, perhaps wishing to conduct a similar study.

Response : Thank you for your suggestion. We used this sample size calculation formula because it is written in a textbook: Clinical epidemiology 4th edition, Huang Yueqin et al, Beijing: People's Medical Publishing House, 2014 (Chinese version). P was taken as the prevalence of Tourette disorder (0.30%), the confidence coefficient (Z) was 95% , and relative error (ϵ) of 15%. This formula has been widely used in other epidemiological studies, and we have added references.

2、 Please check the references again. The Journal is missing for reference 26 and reference 7, the authors and journal are missing.

Response : Thanks a lot for your comment. We had checked the references to make sure they all meet the requirements of the journal. For reference 26, we added the volume, pages and DOI. For reference 26, the authors and journal are listed.

VERSION 3- REVIEW

REVIEWER	Dr. Sarah Jane Nevitt University of York, Centre for Reviews and Dissemination
REVIEW RETURNED	13-Feb-2024

GENERAL COMMENTS	Thank you, all comments well addressed
--

VERSION 3 – AUTHOR RESPONSE